Changes in Coleopteran assemblages over a successional chronosequence in a Mexican tropical dry forest

http://orcid.org/0000-0002-5372-0250 Díaz-Álvarez Edison A. 1
Manrique Cesar 2
Boege Karina 3
http://orcid.org/0000-0003-3862-1024 del-Val Ek 2 ekdelval@iies.unam.mx
1 Instituto de Investigaciones Forestales, Universidad Veracruzana , Xalapa, Veracruz , Mexico
2 Instituto de Investigaciones en Ecosistemas y Sustentabilidad, Universidad Nacional Autónoma de México , Morelia, Michoacan , Mexico
3 Instituto de Ecología, Universidad Nacional Autónoma de México , Mexico City , Mexico
Silva Daniel
Electronic publication date: 2023 Jul 12
Publication date: 2023
Volume: 11
Electronic Location ID: e15712
Received 2023 Feb 22; Accepted 2023 Jun 15
Copyright: © 2023 Díaz-Álvarez et al.
Copyright year: 2023
Copyright holder: Díaz-Álvarez et al.
License: This is an open access article distributed under the terms of the Creative Commons Attribution License, which permits unrestricted use, distribution, reproduction and adaptation in any medium and for any purpose provided that it is properly attributed. For attribution, the original author(s), title, publication source (PeerJ) and either DOI or URL of the article must be cited.
License URL: https://creativecommons.org/licenses/by/4.0/

Keywords: Community ecology, Coleopteran diversity, Ecological succession, Global change, Land-use change, Trophic guilds, Beetles, Tropical dry forest, Agricultural matrix

Funding: MABOTRO. Bases científicas para la conservación restauración y aprovechamiento de ecosistemas CONACYT-SEMARNAT 2002-C01-0597 PAPIIT‐UNAM IN208610 This project was part of the study “MABOTRO. Bases científicas para la conservación restauración y aprovechamiento de ecosistemas” CONACYT-SEMARNAT 2002-C01-0597. Additional funding was provided by PAPIIT‐UNAM, Grant/Award Number: IN208610. The funders had no role in study design, data collection and analysis, decision to publish, or preparation of the manuscript.

==============================
Coleopterans are the most diverse animal group on Earth and constitute good indicators of environmental change. However, little information is available about Coleopteran communities’ responses to disturbance and land-use change. Tropical dry forests have undergone especially extensive anthropogenic impacts in the past decades. This has led to mosaic landscapes consisting of areas of primary forest surrounded by pastures, agricultural fields and secondary forests, which negatively impacts many taxonomic groups. However, such impacts have not been assessed for most arthropod groups. In this work, we compared the abundance, richness and diversity of Coleopteran morphospecies in four different successional stages in a tropical dry forest in western Mexico, to answer the question: How do Coleopteran assemblages associate with vegetation change over the course of forest succession? In addition, we assessed the family composition and trophic guilds for the four successional stages. We found 971 Coleopterans belonging to 107 morphospecies distributed in 28 families. Coleopteran abundance and richness were greatest for pastures than for latter successional stages, and the most abundant family was Chrysomelidae, with 29% of the individuals. Herbivores were the most abundant guild, accounting for 57% of the individuals, followed by predators (22%) and saprophages (21%) beetles. Given the high diversity and richness found throughout the successional chronosequence of the studied tropical dry forest, in order to have the maximum number of species associated with tropical dry forests, large tracts of forest should be preserved so that successional dynamics are able to occur naturally.

Introduction

Land use change is the leading cause of biodiversity loss worldwide (Sala et al., 2000; Betts et al., 2017). In Latin America, the conversion of forest to agricultural, livestock fields, and urban centers has led to a loss of up to 25% of the original forest vegetation area between 2001 and 2015, severely affecting tropical areas of the region (Curtis et al., 2018). In Mexico, the situation is particularly worrying, as it is among the 10 countries with the greatest loss of primary forest worldwide. Between 2001 and 2019, at least 3.99 million hectares of forest cover were lost (Global Forest Watch, 2020), including tropical dry forests, whose alarming 1.4% annual deforestation rate has already caused huge losses for these Mexican ecosystems (Global Forest Watch, 2020).

The tropical dry forest of Chamela-Cuixmala in western Mexico is considered one of the most diverse of its type, with a high percentage of endemic plant species (Trejo & Dirzo, 2002). However, in accordance with the national trend, the forest cover in this region was reduced by at least 80% from the 1970s due to agriculture and cattle ranching, creating a mosaic of forest fragments surrounded by pastures, agricultural fields and secondary forests (Sánchez-Azofeifa et al., 2009). However, many of these fields were later abandoned due to loss of soil fertility (Trejo & Dirzo, 2000, 2002; Flores-Casas & Ortega-Huerta, 2019), providing opportunities for forest succession and reestablishment of tropical dry ecosystems.

Disturbance caused by deforestation has a profound impact on the community structure of ecosystems and on the ecological services they provide (Turner, 2010; Cajaiba et al., 2015). In particular, disturbance can affect the complex relationships between plants and insects (Villa-Galaviz, Boege & del-Val, 2012; Boege et al., 2019). Coleopterans are particularly sensitive because many species are specialist herbivores of particular plant species (Bernays & Chapman, 1994) and are adapted to specific types of vegetation that provide the adequate microclimatic conditions for their survival (Nichols et al., 2007). For example, changes in canopy cover modify light intensity and humidity regimes, causing drastic reductions on Coleopteran populations (Navarrete & Halffter, 2008; Viegas et al., 2014; Cajaiba et al., 2017; Sánchez-Reyes et al., 2019). Despite being the most speciose group of animals on Earth, with up to 400,000 species known to science, Coleoptera, like many other taxonomic groups, are highly vulnerable to global change, particularly to climate and land use change (Dirzo et al., 2014; Elias, 2015; Stork et al., 2015; Stork, 2018).

Coleopterans are adapted to all types of habitats except for the seas and polar regions, and their sensitivity to environmental changes, vast abundance, and broad ecological diversity make them good indicators of the condition of biodiversity in a given territory in the context of global change (Yeates, Bouchard & Monteith, 2002; Cajaiba et al., 2017). While the disturbance and recovery cycle of ecosystems is an important natural component of community ecology, anthropic disturbance has modified this natural process on an unprecedented scale worldwide (Turner, 2010; Chang & Turner, 2019). In addition, the high diversity of the tropical dry forests and their current status as one of the most threatened ecosystems of the world demands a full understanding of its regeneration and resilience (Quesada et al., 2009). It is therefore important to know how animal communities respond to disturbance and recovery throughout the progression of forest succession, which can be explored through the study of successional gradients. In this context, we compared the abundance, richness and diversity of Coleopteran morphospecies among four different successional stages of a tropical dry forest in Mexico to answer the question, how do Coleopteran assemblages associate with vegetation change over the course of forest succession? We also assessed the family composition and trophic guilds for the four successional stages.

Materials and Methods

Study area

The study was conducted in the Chamela-Cuixmala Biosphere Reserve (CCBR, 19°22′–19°39′N, 104°56′–105°10′W), and surrounding areas, located in the municipality of La Huerta in the Pacific coast of Jalisco, Mexico (Fig. 1). The reserve includes an area of 13,142 protected hectares that contains well-preserved mature forest. The annual rainfall for the region averages 795.8 mm but varies greatly from year to year (from 366 to 1,329 mm), and 87% of the annual precipitation occurs between June and October (Maass et al., 2018). Vegetation in this area consists primarily of tropical deciduous forest composed of up to 277 tree species with an average canopy height of 6 m, and semi-deciduous forest established along large streams, where the average canopy height is 10 m (Avila-Cabadilla et al., 2009). The invertebrate inventory is still very limited: 1,877 arthropod species have been described in the reserve, 583 of which are lepidopterans (Pescador-Rubio, Rodríguez-Palafox & Noguera, 2002). There are some studies of particular families of Coleoptera, identifying 308 Cerambicids (Noguera, 2014), 49 Cantaroidea (Zaragoza-Caballero & Ramírez-García, 2009) and 33 Buprestidae (Pérez-Hernández et al., 2023).

Figure 1 Study area.

(A) Mexico and Jalisco highlighted in gray, (B) the municipality of La Huerta indicated in green, and (C) experimental plots for the seven selected sites in the Chamela-Cuixmala reserve and surrounding areas; the colour points indicate the specific sites, and the red line indicates boundary of the Chamela-Cuixmala reserve belonging to the municipality of La Huerta, Jalisco, indicated in yellow. Satellite images were obtained from Google Earth Pro 7.3.

Successional chronosequence

The study included 12 sites that are integrated into the experimental design of the CIECO-UNAM Tropical Forest Management project (MABOTRO; Avila-Cabadilla et al., 2009; Fig. 1). The sites included plots with different ages of succession, but with similar substrate and land use histories, which allowed inference of the successional process over time (Alvarez-Añorve et al., 2012). The chronosequence consisted of four categories: the first three corresponded to sites originally covered by tropical deciduous forest that was cleared for livestock and agricultural activities and later abandoned. These sites were designated according to their successional stages as: pasture, for plots abandoned within the past 2 to 3 years; early successional forest, for plots abandoned 7 to 9 years prior; and intermediate successional forest, for plots abandoned 12 to 16 years prior. The fourth category consisted in plots of mature forest, which had not been disturbed for at least 60 years. The age of abandonment was determined based on interviews with the landowners.

Coleopteran sampling

At each site, one plot of 120 m × 90 m was established and delimited with a fence to exclude livestock and avoid human disturbance. For each plot, four parallel transects of 20 m × 2 m separated by 20 m were established. Along each transect, the Coleopteran community associated with the vegetation was collected four times during the rainy season of 2007, from July to November. The sampling was conducted only during the rainy season because most of trees of this ecosystem have no leaves during the dry period and therefore Coleopteran activity related to aboveground tree biomass is strongly reduced during the dry season.

Sampling in the four successional stages was conducted using sweep nets, in addition, for the three successional categories with shrubs and trees, we carefully examined the leaves and stems of all the trees and shrubs within the transects for direct collection. For sampling in taller trees, three branches of the canopy per tree were selected randomly for direct collection (for further details, see Boege et al., 2019). Coleopteran specimens were identified as morphospecies (Hale et al., 2005) to the highest degree of detail possible, a technique known as “taxonomic sufficiency” (Ellis, 1985) or “lowest practical taxonomic level” (LPT) (e.g., Hanula et al., 2009). In addition, for adult specimens, morphospecies were assigned to a trophic guild (herbivores, predators and saprophages) using insect identification guides, considering the habits known for the family to which they belong (Borror, Triplehorn & Johnson, 1989; White & Peterson, 1998; Eaton & Kaufman, 2007). Collection permit was provided by the Secretaría de Medio Ambiente, Recursos Naturales y Pesca, Subsecretaria de Gestión para la Protección Ambiental (SGPA/DGVS/02005/08).

Data analyses

Coleopteran diversity along the successional gradient was estimated by sample-size and coverage-based integrations of rarefaction and extrapolation of Hill numbers, or the effective number of species (Chao et al., 2014; Hsieh, Ma & Chao, 2016). This method is recommended to compare species diversity across multiple assemblages that differ in sample size (Chao et al., 2014; Ellison, 2010; Hsieh, Ma & Chao, 2016). We utilized Hill numbers, consisting of species richness (q0: which does not consider species abundance); Shannon diversity (q1: which counts species in proportion to their abundances, thus assessing the effective number of common species); and Simpson diversity (q2: which gives greater weight to species evenness than Shannon’s diversity; Chao et al., 2014; Hsieh, Ma & Chao, 2016). We used the R package iNEXT (Hsieh, Ma & Chao, 2016) to produce rarefaction and extrapolation sampling curves for each Hill number. Non-overlapping confidence intervals indicate significant differences among successional stages (Hsieh, Ma & Chao, 2016). Differences in diversity indices between successional stages were analyzed with two-way ANOVAs, considering the diversity estimators or abundance as the response variables, and site and sampling as explanatory variables. To determine the differences among the number of morphospecies at the trophic guilds along the successional gradient, we conducted a one-way ANOVA followed by a Tukey test. The analyses were conducted with R (version 3.6.1; R Core Team, 2019).

β diversity and species assemblages

We calculated the β diversity among the different successional stages using Sorensen’s qualitative measure as the similarity index, which reveals differences between habitats along environmental gradients (Anderson et al., 2011; Socolar et al., 2016). In addition, we conducted a cluster analysis using Jaccard distances to evaluate the similarity in morphospecies composition among successional stages. We conducted these analyses using the vegan and betapart packages in R (version 3.6.1, R Core Team, 2019; Baselga & Orme, 2012; Oksanen et al., 2017).

Results

Abundance, diversity and richness

We collected 971 Coleopteran individuals belonging to 107 morphospecies distributed in 28 families during the 2007 rainy season. The most abundant family was Chrysomelidae, with 29% of the individuals, followed by Curculionidae (15%), Buprestidae (10%), Coccinellidae (5%), Cerambycidae (5%) and Elateridae (5%); the rest of families accounted for the remaining 30%. Coleopteran abundance varied along the successional chronosequence (F(3,8) = 9.36, P = 0.005), with the highest abundance found in pasture plots (Fig. 2).

Figure 2 Abundance of Coleopteran families along a successional chronosequence in the tropical dry forest in Mexico.

The measurements were calculated from the abundance letters indicate significant differences (P ≤ 0.05). Data are shown as the mean ± S.E.

Coleopteran richness (q0) was greater at the early and intermediate stages, followed by the mature forest and pastures, according to the rarefaction curves (Fig. 3). Nevertheless, the sampling effort was not sufficient to detect all the expected Coleopteran species in all successional stages, as the asymptote was only reached for pasture plots. Additionally, rarefaction curves showed that the capture effort required to obtain the total species richness differed among the four stages; the species were generally more abundant in the pasture and rarer in the more mature stages. It is worth mentioning that many morphospecies were represented by one or few individuals in our sampling, corroborating the fact that longer sampling periods are advisable for future studies.

Figure 3 Rarefaction and extrapolation curves.

(A) Species richness (q0), (B) Shannon diversity (q1) and (C) Simpson diversity (q2) of Coleopteran species sampled during 2007 in four successional stages of a tropical dry forest in the Chamela, Jalisco region.

The diversity of common Coleopteran species (i.e., Shannon diversity, q1) and dominant species (i.e., Simpson diversity, q2) differed among the successional stages; early successional plots had the highest diversity for both indices followed by pastures and the intermediate succession plots and finally by mature forest plots, which showed the lowest values for all three indices (Figs. 3A–3C).

When assessing sampling completeness, Simpson (q2) and Shannon (q1) showed high values of completeness (q2: pastures = 0.97, early = 0.85, late = 0.81 and forest = 0.96; q1: pastures = 0.93, early = 0.72, late = 0.69 and forest = 0.71), while Richness (q0) completeness was low in most of the successional stages (q0: pastures = 0.82, early = 0.6, late = 0.59 and forest = 0.47).

Family composition along the successional chronosequence

Chrysomelidae and Curculionidae represented 50% of individuals throughout the whole successional chronosequence. Pastures had the highest number of families (21), and Chrysomelidae represented 32% of the individuals, followed by Curculionidae with 15%, and Coccinellidae with 7%. The early successional stage had 15 families, and Chrysomelidae was represented by 34% of individuals, followed by Curculionidae with 18%, Buprestidae 14%, Elateridae 8%, and Phalacridae 8%. The intermediate stage had 13 families, with Chrysomelidae representing 34% of individuals, followed by Curculionidae with 18%, Lycidae, Elateridae, Coccinellidae, Cerambycidae and Buprestidae with 6% each. For mature forest, 18 families were found, with Chrysomelidae representing 32% of individuals, followed by Curculionidae with 13%, Elateridae 8% and Tenebrionidae, Cerambycidae, Coccinellidae, and Phalacridae had 5% each. The sum of the other families represented the remaining 30% (Fig. 4).

Figure 4 Coleoptearan family abundance.

Abundance of Coleoptera families at each successional stage as the percentage of individuals captured between August and November 2007.

Trophic guilds along the successional gradient

Herbivores had the highest abundance with 57% of the individuals, followed by predators with 22%, and saprophages with 21%. This pattern was observed throughout the chronosequence, except for the intermediate succession stage, where saprophages were absent (Fig. 5). We found differences with respect to the number of morphospecies in each guild through the succession; herbivores were represented by the highest number of morphospecies (P < 0.05).

Figure 5 Coleopteran guilds.

Guilds of the Coleoptera families for each successional stage at the tropical dry forest of Chamela and surroundings.

β diversity and cluster analysis

Coleopteran morphospecies similarity was variable among the successional stages. The Sorensen similarity index showed that pasture and the early stage plots shared the highest number of morphospecies (63%), while mature forest and pastures shared only 37.7% (Table 1). The Sorensen similarity index using the conglomerate technique showed that successional stages aggregated according to successional progression, forming three morphospecies assemblages. The first assemblage included the pasture and early succession stages; the second contained only the intermediate stage with a similarity percentage of 33.7% with respect to the first association; and the third included the mature stage with low similarity 24.3% with respect to the earlier successional stages (Fig. 6).

Table 1 Similarity measure of shared morphospecies (Sorensen qualitative similarity index) by successional stage in the tropical dry forest in the Chamela-Cuixmala region.

Successional stages	Pasture	Early	Intermediate	Mature forest	
Pasture		0.631	0.404	0.377	
Early	0.631		0.504	0.4	
Intermediate	0.404	0.504		0.425	
Mature forest	0.377	0.4	0.425		
Note:

A value of 1.0 indicates that 100% of the morphospecies were shared between the two stages, while 0 indicates no shared morphospecies.

Figure 6 Cluster analysis using Jaccard distance measurement.

The figure shows the similarity in species composition between successional stages based on the composition of Coleopteran morphospecies collected in each of the successional stages.

Discussion

The study of species diversity throughout different habitats contributes to our understanding of ecosystem functioning and allows us to monitor changes that occur in biota under disturbance that results from natural phenomena or human actions (Moreno & Halffter, 2001; Turner, 2010). While disturbance is a key ecological process, it involves the disruption of the ecosystem’s structure at many levels, for example, changing community assemblages (Turner, 2010). Depending on the magnitude of the disturbance, the original species of the ecosystem can be partially or completely replaced by others. In the case of the dry forest of Chamela, we observed a partial replacement of Coleopterans, since the pasture shared some species with the other successional stages.

The way disturbances among forest successional stages influence Coleopteran species assemblages depend on the magnitude of the disturbances (Siqueira Neves et al., 2010; Cajaiba et al., 2017). For example, the Coleopteran species richness and diversity are higher in more recently disturbed habitats than in preserved ones because after perturbation, novel resources promote the arrival of new species, adding to the taxa from the original habitat that were able to persist after disturbance (Sánchez-Reyes et al., 2019). Hence, although specialist species require host plants that are characteristic of mature successional stages, several taxa can be shared among different early successional stages (DeWalt, Maliakal & Denslow, 2003; Cajaiba et al., 2017; Sánchez-Reyes et al., 2019; Faccion et al., 2021), as we found for the pasture and early successional stages of our study. However, how long will the richness and diversity of these modified environments remain high is still a matter of research, in particular, taking into account the projections of larger anthropic environmental alterations in the near future, and with this an unknown impact on Coleopteran communities (Sage, 2019). Some studies have found that disturbed forests or early successional stages can be inundated with exotic or invasive insect species (Gandhi & Herms, 2010; Liebhold et al., 2017), in this study we were not able to identify the origin of morphospecies (i.e., native vs exotic) but none of the Coleopterans found is considered as pest.

Several processes could explain the findings in our study and others of higher diversity in earlier successional stages and species sharing among mature and early successional stages. First, our early successional plots were free of any anthropic intervention for at least three years; for example, livestock and its negative effects on Coleopteran diversity (Cajaiba et al., 2017). Hence, the high species diversity found in early successional plots suggests an important effect of the natural recovery of this taxon. Second, food sources are different among successional stages. Novel food resources in early successional stages, such as flowering herbs, can favor different feeding strategies of many generalist families that are mainly associated with open environments and that can use different plant resources across the landscape. That is, while open environments provide important resources such as flowers that can be used by adult Coleopterans, larval stages of those same species can be found feeding on plant vegetative tissues in the understory of early succession stages (DeWalt, Maliakal & Denslow, 2003; Cajaiba et al., 2017; Faccion et al., 2021). In addition, many Coleopteran families are specialists that require resources that are only available in mature successional stages; this is the case in groups that utilize trees for refuge and food, such as many species of the Cerambycidae and Scarabaeidae (Noguera et al., 2012; Cajaiba et al., 2017), very conspicuous families in our study as discussed below.

Our rarefaction curves showed that our sampling effort was not completely sufficient to characterize the complete Coleopteran fauna, evidenced especially by missing individuals from different families in the mature successional stages. Rarefaction curves suggested that species from the more advanced succession sites were rarer, whereas the pastures presented more common (and thus more easily detectable) species. Therefore, to reach a complete characterization of the tropical dry forest succession it is necessary to continue sampling for a longer time, particularly in the mature forest and including more forest strata. This can be achieved by using additional collecting methods such as pitfall, malaise and flight interception traps.

Interestingly we found that 40% of the Coleopteran species are shared between stages and there is a 20% turnover of the total species for all stages. There were three major associations in terms of species similarity, which grouped assemblages in accordance with the chronosequence, with earlier stages being more similar to each other than to later successional stages. This pattern could be explained by the niche dynamics, in which the specialization of some species for certain resources propitiates the exclusion of others, this pattern is commonly observed in environments with large gradients as those during forest succession (Verberk, 2011). In our case for Coleopterns, the particular resources found at each successional stage determines the presence of a given species; in particular, in mature well conserved forests. Therefore, although there is a significant number of species shared among all stages, no single stage contains the full diversity of the area. Thus, in order to have the maximum number of species associated with tropical dry forests, large areas of forests should be preserved so that successional dynamics are able to occur naturally. Previous studies in tropical dry forest have shown that most Coleopteran herbivores belong to the Chrysomelidae, Curculionidae, Buprestidae and Elateridae families, the last of which is a conspicuous component of Coleoptera fauna in the region (Macedo-Reis et al., 2016; Martínez-Luque, Zurita-García & Zaldívar-Riverón, 2016; Reyes-González et al., 2016; Vega-Badillo et al., 2018). Our results were consistent with these studies, and the predominance of herbivores was due to the high abundance of individuals from these families in our sampling focused on species associated with vegetation. Although we found a large number of herbivores, in part, as a consequence of our sampling method, we also found species from other guilds that are associated with the vegetation, which confirms that changes in the vegetation in our study area do not only affect herbivores, but also, other guilds, including saprophages and predators, as have been observed in other studies (Boege et al., 2019).

The saprophages group in the tropical dry forest contain families such as Nitidulidae, Tenebrionidae, Scarabaeidae, Silphidae, Cupepidae and Ptinidae (Deloya, Madora & Covarrubias, 2013; González-Ramírez, Zaragoza-Caballero & Pérez-Hernández, 2017), which were also present in our study system. Moreover, predators include members of the families Staphylinidae, Carabidae, Coccinellidae, Lampyridae, Lycidae and Cicindelidae (Jiménez-Sánchez, Zaragoza-Caballero & Noguera, 2009; González-Ramírez, Zaragoza-Caballero & Pérez-Hernández, 2017), also found in our sampling. Another conspicuous family in the tropical dry forest is Cerambycidae, a polyphagous group, which is important for the region and other tropical dry forests in Mexico (Chemsak & Noguera, 1993; Noguera et al., 2002, 2009, 2012; González-Ramírez, Zaragoza-Caballero & Pérez-Hernández, 2017). This family, along with the other important families we found in this environment, contribute to ecosystem structure and functioning, in particular, for the energy flow through the trophic web, as many species of these families occur at different levels of the web (Schowalter, 2016). Also, it is worth recalling that herbivores and predators also function as controllers for potentially harmful species in the ecosystem (Schowalter, 2016).

Most successional stages showed a similar compositional pattern in terms of guilds. On the one hand, this is related with our sampling method, but also as a result of the guild assignment at the Family level, based on results reported in other works with greater abundance of herbivores, followed by predators and saprophages (Crowson, 1968; Paulian, 1988; Lawrence & Newton, 1995; Pérez, 1996; Arnett & Thomas, 2000; Arnett et al., 2002). Several works have reported a greater number of guilds in mature successional stages than for the early ones due to wider resource availability for different groups (Tscharntke et al., 2002; Lassau et al., 2005; Solervicens, 1995). For example, the amount of detritus at later than earlier successional stages due to higher production of leaf litter (Barberena-Arias & Aide, 2003), which is a resource used by both predators and saprophages. The Coleopteran communities in the litter and soil are responsible for different ecosystem services for example, saprophages contribute to the release of nutrients stored in dead matter, which in turn are important for nutrient cycling in these ecosystems (Stork & Grimbacher, 2006; Slade, Mann & Lewis, 2011; Barretto, Cultid-Medina & Escobar, 2018). Hence, further studies should consider the effect of disturbance on the diversity, richness and ecosystem services on saprophages and other guilds such as detritivores.

The high diversity of plants that contribute to the formation of different forest strata provides heterogeneity of habitats and resources for different Coleoptera groups (Barberena-Arias & Aide, 2003; Taboada et al., 2010; Sánchez-Reyes et al., 2019). As is the case for succession, in which the heterogeneity at each successional stage has an important role for Coleopteran assemblages, as our study describes. This is also likely to occur for other groups of insects such as Lepidoptera, Hemiptera and Orthoptera, which together with Coleoptera have high abundance in the tropical dry forests and contribute with the complex ecological interactions among different guilds, for example, as available prey for predators from different groups of animals (Barberena-Arias & Aide, 2003; Sánchez-Reyes et al., 2019; Larsson Ekström, Bergmark & Hekkala, 2021). Such interactions are not only relevant in the case of adult insects, but for other life-stages, so, considering we only studied adult Coleopterans, we are missing information about larval diversity and their functional role throughout the chronosequence. Therefore, further studies should address other life-stages, not only for Coleoptera, but also for other insect groups to complement our research and understand the complex ecological interaction in modified environments, both between animals and plants, and between insects from different groups.

Conclusions

Our study makes an important contribution to understanding the community assembly of one of the most threatened ecosystems of the world. Considering that every successional stage is important for the Coleopterans in this tropical ecosystem, and given the importance of Coleopterans for ecosystems functioning, urgent actions are needed to conserve this group of animals and the associated ecosystems. Our results suggest that an effective approach to forest conservation should consider a mosaic of successional stages that maximizes the conservation of Coleopteran diversity. This type of study represents the basis for biodiversity conservation programs, which should prioritize an understanding of community dynamics throughout succession to facilitate the planning and management of areas that are actively undergoing regeneration processes, including many protected natural areas.

Supplemental Information

Supplemental Information 1 Morphospecies abundance per plot.

Each data point represents abundance of each morphospecies per sampling per plot. These morphospecies abundances were used to calculate the Hill numbers, including the assigned family and trophic guild.

Click here for additional data file.

We especially thank Ruben Pérez-Ishiwara for all of the logistical and administrative support. The manuscript was revised by the professional English translator Lynna Kiere.

Additional Information and Declarations

Competing Interests

Author Contributions

Field Study Permissions

Data Availability

The authors declare that they have no competing interests.

Edison A. Díaz-Álvarez analyzed the data, prepared figures and/or tables, authored or reviewed drafts of the article, and approved the final draft.

Cesar Manrique performed the experiments, authored or reviewed drafts of the article, and approved the final draft.

Karina Boege conceived and designed the experiments, authored or reviewed drafts of the article, and approved the final draft.

Ek del-Val conceived and designed the experiments, performed the experiments, analyzed the data, authored or reviewed drafts of the article, and approved the final draft.

The following information was supplied relating to field study approvals (i.e., approving body and any reference numbers):

Collection permit was provided by the Subsecretaria de Gestión para la Protección Ambiental (SGPA/DGVS/02005/08).

The following information was supplied regarding data availability:

The raw measurements are available in the Supplemental File.

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
