# Peer review of "Changes in Coleopteran assemblages over a successional chronosequence in a Mexican tropical dry forest"

_PeerJ, doi:10.7717/peerj.15712_

## Round 0.1 · original submission · Major Revisions

Dear Dr. del-Val,

In this first review, your manuscript was evaluated by two independent reviewers. Both reviewers indicated that your manuscript has merits and may be published in PeerJ after a major review.

Please take a close look at what the reviewers raised and make the necessary changes to your text. After that, please resubmit your manuscript along with a rebuttal letter informing the changes you implemented and justifying those you did not.

Please also note that one of the reviewers made extensive suggestions for improving your manuscript. I agree with all the issues both reviewers raised.

Best regards,
Daniel Silva

Reviewer 1 ·

Basic reporting

The quality of the English is generally fine, though there are still a few spelling or grammatical errors. Other than that the structure of the manuscript is fine.

Experimental design

The aims of the paper and the gap it could potentially fill are all relevant. The methods are well detailed, though I have reservations that the methodology the authors employed was appropriate for the questions they wished to answer.
• Effective collecting effort and techniques varied among habitat types (sweep netting in pasture, inspecting branches in forest), which poses issues when making comparisons in diversity and composition. The pasture was effectively very well sampled by sweep netting, hence the high abundance and diversity of beetles, and the asymptote was reached for species richness (which almost never happens). By contrast, the more forested sites were under sampled. Comparisons in diversity and especially composition therefore are likely inaccurate since you are comparing a very well sampled habitat with relatively poorly sampled habitats. A greater sampling effort in the forests would no doubt have uncovered many species that were also collected in the other habitat types, thus reducing dissimilarity among sites, and increasing diversity within habitat types. A more consistent approach would have been to set up identical traps (e.g. pitfall and/ flight intercept traps) in each habitat for direct comparisons. It is also difficult to be consistent in sampling effort using hand-collecting techniques, even if only performed by one person, and spot sampling can be influenced greatly by current or preceding weather. If the authors have the option of continuing sampling at these sites, I recommend utilising traps (which are excellent for between-site comparisons) rather than hand-collecting techniques (which are most appropriate for between-host plant comparisons). Traps would also be unaffected by leaf loss during the dry season, allowing for an examination of seasonal patterns.

• The collecting techniques were also bias towards herbivorous species as they specifically targeted above-ground vegetation. If the purpose of the study is to assess biodiversity, then other trapping techniques would have been more appropriate for collecting a wider variety of species. The guild composition analyses are therefore somewhat compromised. The number of species and individuals collected was also quite low, and certainly only represents a small portion of the regional species richness of beetles. Traps would more thoroughly sample the local beetle community.

Validity of the findings

The issues with collecting techniques mean that the major conclusion the authors draw from their data, that a mosaic of habitat types should be preserved to maintain high beetle diversity, is not well-supported. It may well be true, but the bias towards pasture assemblages and herbivores means that comparisons are weak and conclusions only tentative. Herbivores are the most likely group to be abundant and diverse across the whole chronosequence. Trapping may well have revealed a diverse assemblage of dead-wood associated species in the more mature forested sites, for example. I suggest the authors acknowledge the short-comings of their approach in the text and present their results within the context of those caveats.

Additional comments

Minor points in the text:
• Line 71: There are about 400,000 described beetle species I believe. Current estimates suggest more than 1 million beetle species, but these are not all known to science.

• Line 102-103: Be consistent in the family names here. What are Cantaroidea? Is this supposed to be Cantharidae? Or should it be Elateroidea as the reference suggests?

• Results: Could we have a list of species names of the beetles collected? Perhaps for the supplementary materials? The current file just has a series of morphospecies names that do not indicate what they are.

• Line 186, 190, and 192: Coccinellidae.

• Line 192: Alleculidae is a subfamily of Tenebrionidae (Alleculinae).

• Line 225: These original forest species may not persist indefinitely though. They may simply take a while to die off.

• Discussion: I’m not sure I fully agree that the entire successional gradient should be preserved. A forest left to its own devices and free of human interference will still suffer disturbances (landslides, fires, floods, outbreaks of disease, windthrow, etc.), which will then lead to succession and recovery. I think it would be more accurate to state that large tracts of forest should be preserved so that successional dynamics are able to occur naturally.

• Lines 263-273: I suggest deleting these comparisons to other studies. The major difference in diversity among these studies is no doubt sampling effort rather than any underlying difference in diversity. Personally I have collected over 200 beetle species from 44 families from an industrial seaport in a major city. I highly doubt my study site, which was basically a concrete wasteland, was more diverse than any cited here. I simply used a variety of traps (including light traps) and collected a lot of specimens (~20,000).

• Lines 274-290: As mentioned already, I think a major reason you found so many herbivores compared to other guilds is because you used hand-collecting techniques and only sampled foliage. This paragraph is therefore largely speculative and should either be deleted or highly modified.

• Line 281: Cupedidae.

• Line 284: Cicindelidae.

• General: there are several instances where the authors make a generic statement about the contribution of various beetle groups to ecosystem services or functions, but there is a lack of any explanation or detail as to what they may do (see for example, lines 289-290, and 301-302). I suggest backing up the generic statements with specific examples.

• Figure 4: There are several errors in this figure. Alleculidae and Lycidae are the same colour. Alleculidae should be included under Tenebrionidae. Bruchidae should be under Chrysomelidae as Bruchinae. Coccinellidae, not Coccinelidae.

·

Basic reporting

The manuscript is generally clear and well-structured. The introduction provides grounding in relevant literature. I have a few minor suggestions for clarity:

line 51: is that 1.4% annual rate of loss just for Mexico, or for tropical dry forests worldwide? Please clarify.

line 132 and Fig. 5 legend: use parallel construction: "herbivores, predators, and saprophages" or "hebivorous, predaceous, and saprophagous"

line 133: family should not be capitalized

line 151: to determine differences in what response variable-- abundance?

line 288: the sentence about Cerambycidae is not very informative; I recommend rewording to describe the pattern along the chronosequence.

Fig. 1. I recommend color-coding the sites by age category.

The raw data are fairly self-explanatory, but I recommend giving the actual dates instead of (or as well as) sample date 1, 2, 3, 4. So that all of the analyses can be replicated from the raw data, it would also be necessary to specify which family and guild each of the morphospecies belongs to.

Experimental design

It is refreshing to see a study that treats biodiversity data in robust ways, using Hill numbers to compare the diversity of different plots at an equivalent amount of individuals. I do have some suggestions for improving the presentation of these data.

1). Make sure that the descriptions of the methods line up between the text, figure captions, and raw data. All methods should be described clearly in the Methods section, presented in the Results, and placed into context in the Discussion. There were a number of minor discrepancies, listed below:

line 121: the raw data shows four sample periods, but the text says five. Please clarify.

line 254: this is the first (and only) mention of the Wilson and Schmida index. If you plan to use this index, it should be included in the methods and results.

Fig. 2. The caption says families but the y-axis says individuals. Please clarify. (I think it must be individuals-- on line 34 you mention that 28 families were found across the entire data set, and the first bar on the figure is higher than that.)

2) Make sure that you cite the appropriate sources for the software and packages you are using.

line 140: you should cite Chao et al. (2014) for the theoretical basis of using Hill numbers.

line 146: you should cite Hsieh et al. (2016) instead of R Development Core Team for the iNEXT package.

3) In considering how complete your sampling was, it would be advisable to examine the sample completeness curve in the iNEXT output (Figs. 1b and 2b in Hsieh et al. 2016). You mention sample completeness in lines 174 and 276, but it is not formally assessed in your paper.


line 153: Why not use a recent release of R? The current version is 4.2.2.

Validity of the findings

The findings are clearly stated and link well with the research questions. I do have a few comments:

1) In the raw data, some of your samples are quite low in diversity (0 or 1 individuals). It would be good to explicitly acknowledge this in your discussion. As a botanist, I'm not as well-versed in the insect literature, so I don't know how this compares to other Coleoptera studies. It struck me as very low, though, and perhaps worth commenting on at greater length.

2) The assumption in line 261 is that all these species are native to the region and are species of potential conservation concern. Is it possible that some of the diversity observed in the pasture and early successional plots comes from non-native species that live on non-native pasture plants? This seems worth exploring, or at least mentioning, in the discussion.

Additional comments

Overall, this study adds to our knowledge of a relatively understudied taxon in a diverse, threatened ecosystem. I commend the authors for the effort that went into collecting these data.

---

## Round 0.2 · accepted · Accept

Dear Dr. del-Val,

After the improvements you made to your manuscript based on the suggestions provided by both reviewers, I now believe your text is suitable for publication in PeerJ. Congratulations on your hard work to improve this study!

Sincerely, Daniel SIlva.